# `CrossSpectra`: Exploiting Cross-Layer Smoothness for Parameter-Efficient Fine-Tuning

**Yifei Zhang**[1,2]*, **Hao Zhu**[4]*, **Junhao Dong**[2], **Haoran Shi**[2], **Ziqiao Meng**[3],
**Piotr Koniusz**[4,5,6]† and **Han Yu**[2]†

[1]School of Computer Science, Northwestern Polytechnical University;
[2]CCDS, Nanyang Technological University; [3]National University of Singapore;
[4]Data61♥CSIRO; [5]University of New South Wales; [6]Australian National University
yifeiacc@gmail.com

## Abstract

Parameter-efficient fine-tuning (PEFT) is essential for adapting large foundation models without excessive storage cost. However, current approaches such as LoRA treat each layer's adaptation independently, overlooking correlations across layers. This independence causes the number of trainable parameters to grow linearly with model depth. We provide theoretical and empirical evidence that skip connections in transformers create smooth gradient propagation across layers. This smoothness leads to weight adaptations that concentrate most of their energy in low-frequency spectral components, especially along the layer dimension. Empirical analysis confirms this effect, showing that most of adaptation energy lies in low frequencies. Building on this insight, we propose `CrossSpectra`, which parameterizes all attention-weight adaptations $(\boldsymbol{Q}, \boldsymbol{K}, \boldsymbol{V})$ across layers as a single 3D tensor and represents them with sparse spectral coefficients $(\kappa_1, \kappa_2)$. Using $\kappa_1$ non-zero coefficients within each layer's frequency space and truncating to $\kappa_2$ frequencies across layers, `CrossSpectra` requires $\mathcal{O}(\kappa_1 \kappa_2)$ parameters instead of LoRA's $\mathcal{O}(Lrd)$, where $L$ is the number of layers and $r$ is the rank. Across natural language and vision benchmarks, `CrossSpectra` matches or surpasses baseline performance while using fewer parameters than LoRA, achieving only $0.36\%$ of LoRA's parameter count when fine-tuning LLaMA-7B on instruction-following tasks. These results show that exploiting the **architectural smoothness of transformers** through spectral analysis yields major efficiency gains in PEFT.

## 1 Introduction

Large Language models (LLMs) have demonstrated exceptional capabilities across domains ranging from natural language processing [Liu et al., 2019, He et al., 2020, Radford et al., 2019, Brown et al., 2020] to computer vision [Liu et al., 2023b,a, Singh et al., 2022, Zhang et al., 2024c] to multi-modal problems [Radford et al., 2021, Dong et al., 2025a,b,c,d, Zhang et al., 2025a,b]) and other domains [Fan et al., 2025, Zhang et al., 2024a, Guo et al., 2024]. However, with model parameters now reaching hundreds of billions or even trillions, adapting these models to specific downstream tasks through conventional fine-tuning has become increasingly impractical. Each customized model typically requires storing as many parameters as the original model [Qiu et al., 2020, Raffel et al., 2020a], leading to substantial storage and deployment challenges as customization needs expand.

Parameter-efficient fine-tuning (PEFT) methods have emerged as promising solutions to this challenge Yang et al. [2024]. These approaches adapt pre-trained models using only a small subset of

---

*Equal Contribution.
†Corresponding Authors.

39th Conference on Neural Information Processing Systems (NeurIPS 2025).

trainable parameters, significantly reducing storage requirements while maintaining performance. Among these methods, Low-Rank Adaptation (LoRA) Hu et al. [2021a] and its variants [Liu et al., 2024, Meng et al., 2024, Wang et al., 2024b, Kalajdzievski, 2023, Si et al., 2024, Zhong et al., 2024, Wang et al., 2024a, Ni et al., 2024]) have gained widespread adoption by representing weight changes through low-rank matrices, achieving impressive results with a fraction of the parameters required for full fine-tuning. Despite these advances, current PEFT methods face a fundamental limitation: they treat each layer's adaptation independently, overlooking potential structural relationships across layers. This independence assumption results in parameter counts that scale linearly with model depth, limiting efficiency gains for increasingly deeper architectures.

A fundamental insight from transformer architecture analysis illuminates a path toward more efficient adaptation. Skip connections, widely used in architectures including ResNets [He et al., 2016] and Transformers [Vaswani et al., 2017], facilitate stable and smooth gradient propagation through depth. This architectural property, combined with the natural spectral bias of neural networks [Rahaman et al., 2019], suggests that weight adaptations during fine-tuning should exhibit exploitable smoothness patterns, particularly across the layer dimension. Empirical analysis confirms this intuition. Figure 1 demonstrates that attention weight adaptations exhibit dramatic spectral concentration across layers, with nearly 70% of adaptation energy concentrated in low-frequency components. This observation motivates a spectral approach to parameter-efficient fine-tuning.

We formalize this intuition through theoretical analysis showing that attention weight adaptations in transformers naturally concentrate in low-frequency components. Our key theoretical findings are: (1) skip connections create Lipschitz-continuous gradient fields across layers, ensuring similar gradients in adjacent layers; (2) these smooth gradients accumulate into smooth weight adaptations during fine-tuning; and (3) the resulting spectral structure shows strongest decay in the cross-layer dimension, enabling aggressive frequency truncation. In this paper, we propose `CrossSpectra`, a novel PEFT method that exploits these cross-layer spectral properties. Instead of parameterizing each layer's adaptations independently, we construct a unified 3D tensor representation of all attention (Query, Key, Value) weight changes and decompose it in the frequency domain. By working directly with spectral coefficients and leveraging 3D inverse FFT for efficient computation, we achieve parameter reductions that scale sub-linearly with model dimensions. Our key contributions are:

i. We establish a theoretical framework connecting skip connection-induced gradient smoothness to spectral properties of weight adaptations, providing a principled foundation for cross-layer parameter sharing in attention mechanisms.

ii. We introduce `CrossSpectra`, a unified tensor formulation that represents all QKV adaptations across layers using sparse spectral decomposition with $\kappa_1$ coefficients per layer and $\kappa_2$ cross-layer frequencies, achieving $\mathcal{O}(\kappa_1 \kappa_2)$ parameter complexity.

iii. We demonstrate that our approach achieves 275× parameter reduction compared to LoRA and 5,250× compared to full fine-tuning, requiring only 8 KB of memory versus LoRA's 2.2 MB for typical transformer configurations.

iv. Through extensive experiments across natural language understanding, instruction tuning, and image classification tasks, we show that `CrossSpectra` matches or exceeds baseline performance while using a fraction of the parameters.

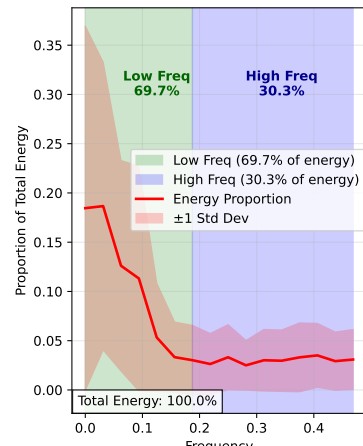

Figure 1: Distribution of spectral energy. Most of adaptation energy concentrates in low frequencies, confirming that skip connections induce smooth, low-frequency dominated weight changes across layer.

## 2 Related Work

**Parameter-efficient Fine-tuning (PEFT) Methods.** As foundation models have grown in size and computational requirements, efficiently adapting them has become essential. PEFT methods fall into two primary categories: non-weight-based [He et al., 2021, Rebuffi et al., 2017, Lester et al.,

2021, Gao et al., 2020, Li and Liang, 2021] and weight-based approaches [Zhu et al., 2025, Hu et al., 2021b, Zhang et al., 2025c, 2024b]. Weight-based methods directly learn modifications to pre-trained weights. LoRA [Hu et al., 2021b] represents weight changes through low-rank matrix decomposition. LoRA has gained widespread adoption due to its simplicity, effectiveness, and ability to merge adapted weights during inference to avoid latency increases. Several extensions have been proposed, including AdaLoRA [Zhang et al., 2023], which adaptively allocates parameter budgets across weight matrices, and FourierFT [Gao et al., 2024] and BiLoRA [Zhu et al., 2025] learn in frequency domain. PACE [Ni et al., 2024] uses noise modulation for generalization guarantees. Despite these advances, all existing methods treat each layer's adaptation independently, failing to exploit potential structural relationships across layers. This independent treatment results in parameter counts that scale linearly with model depth, limiting efficiency gains for increasingly deeper architectures.

**Skip Connections in Deep Learning.** Skip connections, first popularized in ResNet [He et al., 2016] and now fundamental to modern architectures like Transformers [Vaswani et al., 2017], allow information to bypass one or more layers by adding the input of a layer block to its output. Their empirical success in enabling training of very deep networks has been well-documented, but the theoretical understanding of their properties continues to evolve. A pivotal insight came from Chen et al. [2018], who established connections between networks with skip connections and ordinary differential equations (ODEs). This perspective views ResNets as discretizations of continuous dynamic systems, where each layer represents a step in a numerical ODE solver. The key implication is that skip connections induce smoothness in representations between consecutive layers, constraining how drastically the network can transform its inputs at each step. Further work has explored the spectral properties of networks with skip connections. Fourier-based analyses [Rahaman et al., 2019, Tancik et al., 2020] have shown that these networks tend to prioritize low-frequency components, which correspond to smoother transformations. This frequency-domain perspective provides additional evidence for the smoothness-inducing nature of skip connections. Several studies have also investigated the optimization advantages provided by skip connections. The gradient stability properties of these architectures [Miyato et al., 2018] explain their trainability at extreme depths, where traditional networks struggle with vanishing or exploding gradients. However, despite these theoretical advances, the implications of skip connection-induced smoothness for parameter-efficient adaptation remain unexplored. Our work bridges this gap by connecting the smoothness properties of representations to the structure of weight adaptations, enabling more efficient parameter sharing across layers.

## 3 Theoretical Foundation for `CrossSpectra`

We develop a comprehensive theoretical framework explaining why neural network adaptations exhibit exploitable spectral structure across layers. Our analysis focuses on attention mechanisms in transformer architectures, revealing how skip connections induce smooth gradient fields that manifest as low-frequency patterns in weight updates. This spectral structure enables dramatic parameter reduction through frequency-domain methods. Our theoretical analysis yields three fundamental insights that directly inform `CrossSpectra`'s design:

i. **Cross-layer gradient smoothness**: Skip connections create smooth gradient propagation across transformer layers, with attention weights (Query, Key, Values) exhibiting particularly strong cross-layer correlation.

ii. **Dimension-specific spectral decay:** The spectral energy of adaptation patterns decays at different rates across dimensions; the decay is strongest along the layer axis, while spatial dimensions within each layer show a slower attenuation of high-frequency components.

iii. **Joint QKV structure**: All attention matrices ($Q$, $K$, $V$) can be efficiently represented in a unified spectral framework due to their shared architectural context.

These insights motivate our unified spectral approach, where we exploit cross-layer patterns through joint parameterization of all QKV matrices.

### 3.1 Gradient Structure in Transformers with Skip Connections

To understand the spectral properties of attention adaptations, we first analyze how skip connections shape gradient flow through transformer architectures.

**Architecture Setup.** Consider a transformer with $L$ layers. Each layer $l \in \{1, \ldots, L\}$ contains multi-head attention with weight matrices: Query $\boldsymbol{W}_l^Q \in \mathbb{R}^{d \times d}$, Key $\boldsymbol{W}_l^K \in \mathbb{R}^{d \times d}$, and Value $\boldsymbol{W}_l^V \in \mathbb{R}^{d \times d}$. The attention mechanism computes:

$$\text{Attention}_l(\boldsymbol{H}_{l-1}) = \text{softmax}\left(\frac{\boldsymbol{H}_{l-1}\boldsymbol{W}_l^Q(\boldsymbol{H}_{l-1}\boldsymbol{W}_l^K)^\top}{\sqrt{d}}\right)\boldsymbol{H}_{l-1}\boldsymbol{W}_l^V. \tag{1}$$

Crucially, skip connections define the layer-wise evolution of hidden representations:

$$\boldsymbol{H}_l = \boldsymbol{H}_{l-1} + \text{MultiHeadAttention}_l(\boldsymbol{H}_{l-1}) + \text{FFN}_l(\boldsymbol{H}_{l-1}). \tag{2}$$

This residual structure is key to understanding gradient smoothness—it ensures that gradient information flows directly across layers while being modulated by local computations.

**Cross-Layer Gradient Analysis.** To quantify gradient behavior across layers, we examine the gradient structure at each layer:

**Definition 3.1** (Layer-wise Gradient). *For a loss function $\mathcal{L}$ and attention matrices $\{\boldsymbol{W}_l^M\}_{l=1}^L$ where $M \in \{Q, K, V\}$, the gradient at layer $l$ is:*

$$\boldsymbol{G}_l^M = \frac{\partial \mathcal{L}}{\partial \boldsymbol{W}_l^M}, \tag{3}$$

*where $\boldsymbol{G}_l^M \in \mathbb{R}^{d \times d}$ has the same shape as $\boldsymbol{W}_l^M$.*

Skip connections promote smooth gradient propagation across layers. Our first key result formalizes this effect under the standard residual–ODE scaling:

**Theorem 3.2** (Gradient Smoothness for Attention Weights). *Under residual scaling $\boldsymbol{H}_l = \boldsymbol{H}_{l-1} + \frac{1}{L}\boldsymbol{R}_l(\boldsymbol{H}_{l-1}; \boldsymbol{W}_l)$ and bounded Jacobians of $\boldsymbol{R}_l$, the gradients of attention weights satisfy*

$$\|\boldsymbol{G}_{l+1}^M - \boldsymbol{G}_l^M\|_F \leq \frac{C_M}{L}, \tag{4}$$

*where $C_M$ is a depth-independent constant.*

Thus, adjacent layers exhibit gradually varying gradients, and skip connections ensure this variation decays with depth, avoiding the abrupt changes typical of purely feed-forward architectures. Intuitively, the residual scaling $\frac{1}{L}$ constrains each layer's transformation to be a small perturbation of its predecessor, so gradient changes accumulate continuously along depth rather than discretely.

## 3.2 From Gradient Smoothness to Spectral Properties

The gradient smoothness directly translates to spectral properties of weight adaptations during fine-tuning. When gradients vary smoothly across layers, the accumulated weight changes exhibit low-frequency dominance in the layer dimension.

**Adaptation Accumulation Process.** During fine-tuning over $T$ steps with learning rate $\eta$, attention weights accumulate updates:

$$\Delta \boldsymbol{W}_l^M = -\eta \sum_{t=1}^{T} \nabla_{\boldsymbol{W}_l^M} \mathcal{L}^{(t)}, \quad M \in \{Q, K, V\}. \tag{5}$$

To analyze the spectral structure, we organize all weight adaptations into a 3D tensor and examine its frequency characteristics.

**Theorem 3.3** (Spectral Concentration for Attention Matrices). *For attention adaptations $\{\Delta \boldsymbol{W}_l^M\}_{l=1}^L$ where $M \in \{Q, K, V\}$, define the 3D adaptation tensor $\boldsymbol{W}^M \in \mathbb{R}^{d \times d \times L}$ with $[\boldsymbol{W}^M]_{:,:,l} = \Delta \boldsymbol{W}_l^M$. Let $\hat{\boldsymbol{W}}^M(n_1, n_2, n_3)$ denote its 3D discrete Fourier transform, where $n_1 \in \{0, ..., d-1\}$ is frequency index for input dimension, $n_2 \in \{0, ..., d-1\}$ is frequency index for output dimension, and $n_3 \in \{0, ..., L-1\}$ is frequency index across layers. The Fourier coefficients exhibit dimension-specific decay with frequency:*

$$|\hat{\boldsymbol{W}}^M(n_1, n_2, n_3)| \leq \frac{C_M}{(n_1+1)^{\beta_{1,M}} \cdot (n_2+1)^{\beta_{2,M}} \cdot (n_3+1)^{\beta_{3,M}}}, \tag{6}$$

*where the decay rates satisfy: $\beta_{3,M} > \beta_{1,M}, \beta_{2,M}$ for all $M \in \{Q, K, V\}$.*

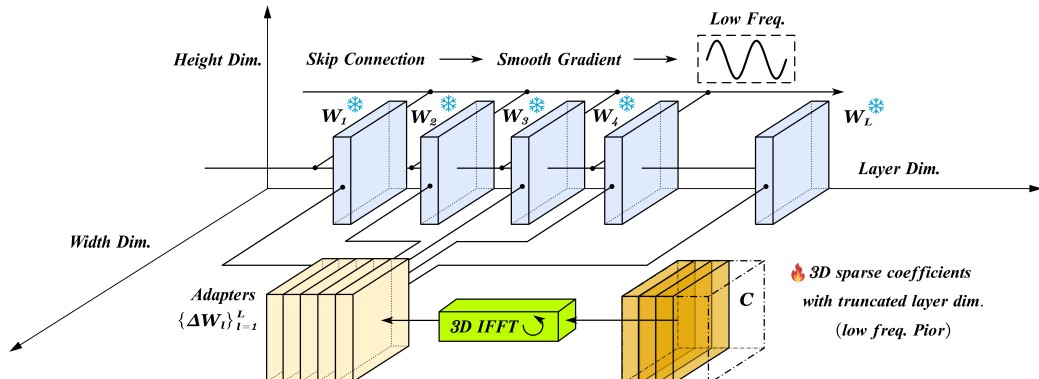

Figure 2: **The overall model**: Skip connections create smooth gradient flow across transformer layers , resulting in weight adaptations that are dominated by low-frequency patterns. `CrossSpectra` exploits this by representing all layers' adaptations as a 3D tensor and using sparse spectral coefficients via 3D inverse FFT, with truncation in the cross-layer frequency dimension.

The stronger decay in the cross-layer dimension ($\beta_{3,M} > \beta_{1,M}, \beta_{2,M}$) implies that adaptations are dominated by low-frequency patterns across layers. This is a direct consequence of the gradient smoothness induced by skip connections—smooth gradients accumulate into smooth adaptations.

# 4   `CrossSpectra`: From Spectral Properties to Efficient Adaptation

Building on our theoretical analysis, we now present `CrossSpectra`, a parameter-efficient fine-tuning method that directly exploits the cross-layer spectral structure in attention weight adaptations. As illustrated in Figure 2, skip connections induce smooth gradient propagation, which leads to low-frequency dominance in the learned adaptations.

**From Theory to Design.** Our theoretical findings directly inform three key design principles for `CrossSpectra`:

  i. **Unified 3D representation**: Since gradients vary smoothly across layers, we parameterize all QKV matrices jointly rather than treating each layer independently.

 ii. **Fourier decomposition with selective truncation**: The strongest spectral decay occurs in the cross-layer dimension ($\beta_3 > \beta_1, \beta_2$), so we aggressively truncate high frequencies in this dimension while preserving more frequencies within layers.

iii. **Implicit spectral regularization**: Frequency-domain parameterization naturally enforces the smoothness constraints revealed by our theoretical analysis.

These principles guide our implementation choices in the following subsections.

## 4.1   Unified QKV Tensor Representation

Traditional parameter-efficient methods decompose each layer's adaptations independently. For example, LoRA parameterizes:

$$\Delta \boldsymbol{W}_l^M = \boldsymbol{A}_l^M \boldsymbol{B}_l^M, \quad \boldsymbol{A}_l^M \in \mathbb{R}^{d \times r}, \boldsymbol{B}_l^M \in \mathbb{R}^{r \times d}. \tag{7}$$

This layer-wise approach cannot exploit the cross-layer patterns identified in our theoretical analysis.

`CrossSpectra` constructs a unified 3D tensor that stacks all QKV matrices across layers:

$$\boldsymbol{T}_{QKV} \in \mathbb{R}^{d \times d \times 3L}, \tag{8}$$

where the dimensions represent $d$ for both input and output dimensions (for simplicity), and $3L$ is the extended layer dimension containing all Q, K, V matrices across $L$ layers. This unified representation enables us to capture cross-layer patterns in a single spectral decomposition, dramatically reducing parameters compared to layer-wise methods.

## 4.2 Spectral Decomposition via 3D Fourier Transform

We decompose the unified QKV tensor using 3D inverse Fourier transform:

$$\boldsymbol{T}_{QKV} = \text{iFFT3D}(\boldsymbol{C}). \tag{9}$$

The coefficient tensor $\boldsymbol{C} \in \mathbb{C}^{d \times d \times 3L}$ is highly sparse, with non-zero entries denoted by the index set $\Omega$. We control this sparsity through two key parameters: $\kappa_1 = |\Omega|/(3L)$ is number of non-zero coefficients sampled within each $d \times d$ layer's frequency space. $\kappa_2$ is number of frequencies retained in the cross-layer dimension ($\kappa_2 \ll 3L$). This parameterization yields a total parameter count of $|\Omega| = \kappa_1 \cdot \kappa_2$, resulting in a sparsity ratio of $|\Omega|/(d^2 \cdot 3L)$. For typical transformer configurations, we achieve dramatic parameter reduction by setting $|\Omega| \approx 0.1\%$ of the full tensor size. The key insight from our theory is that cross-layer spectral decay is strongest ($\beta_3 > \beta_1, \beta_2$). We exploit this by aggressive truncation in the layer dimension ($\kappa_2 \ll 3L$) while employing sparse sampling within each layer ($\kappa_1 \ll d^2$), targeting the most important frequency components.

**Complex Fourier Bases with Real-Valued Output.** We use complex-valued inverse 3D Fourier transforms for computational efficiency. To ensure that the reconstructed tensor $\boldsymbol{T}_{QKV} = \text{iFFT3D}(\boldsymbol{C})$ is real-valued, the sparse coefficient tensor $\boldsymbol{C}$ must satisfy discrete *Hermitian symmetry*:

$$[\boldsymbol{C}]_{u,v,w} = \overline{[\boldsymbol{C}]}_{(-u) \bmod d, \ (-v) \bmod d, \ (-w) \bmod (3L)}, \quad \forall (u, v, w) \in \Omega. \tag{10}$$

Here $(u, v, w)$ are frequency indices along the input, output, and layer dimensions, and $\overline{(\cdot)}$ denotes complex conjugation. This condition guarantees that $\text{iFFT3D}(\boldsymbol{C})$ yields a real-number tensor. Consequently, only one half-space of frequency coefficients needs to be parameterized independently, effectively halving the number of learnable spectral parameters and reducing storage cost.

**Efficient Implementation via 3D iFFT.** `CrossSpectra` leverages highly optimized FFT implementations for computational efficiency. The core operations are:

i. **Forward Pass** (Algorithm 1): We first convert the sparse spectral coefficients $\boldsymbol{C}$ to the spatial domain using 3D inverse FFT, yielding the full adaptation tensor $\boldsymbol{T}_{QKV}$. This operation is performed only once per forward pass. We then extract the appropriate slice for each layer and attention matrix $(\boldsymbol{Q}, \boldsymbol{K}, \boldsymbol{V})$ and add it to the pre-trained weights.

ii. **Backward Pass** (Algorithm 2): We collect all gradient updates into a single tensor, transform it into the frequency domain using 3D FFT, and keep only the gradients corresponding to the sparse indices in $\Omega$. This effectively projects the gradient into our low-dimensional spectral subspace.

This implementation ensures that computation scales with the full tensor dimensions only for the FFT operations, while parameter storage and updates remain proportional to $|\Omega|$.

**Implicit Gradient Regularization.** Algorithm 2 reveals that the backward pass acts as a gradient filter. When we compute $\nabla_{\boldsymbol{C}} = \text{FFT3D}(\nabla_{\boldsymbol{T}_{QKV}})$ and retain only $\kappa_1$ sparse coefficients within each layer's frequency space and $\kappa_2$ frequencies across layers, we project gradients into a subspace with low-frequency basis. This filtering prevents the accumulation of rapid variations, with the strongest regularization occurring in the cross-layer dimension where truncation is the most aggressive. The optimization process is thus biased toward discovering smooth adaptation patterns that align with the natural spectral structure revealed by our theoretical analysis.

## 4.3 Complexity Analysis

Table 1 provides a comprehensive comparison of memory requirements and computational costs.

| Method | Memory | Forward Pass | Backward Pass |
|---|---|---|---|
| Full Fine-tuning | $\mathcal{O}(3Ld^2)$ | $\mathcal{O}(Lnd^2)$ | $\mathcal{O}(Lnd^2)$ |
| LoRA (rank $r$) | $\mathcal{O}(6Lrd)$ | $\mathcal{O}(2Lnrd)$ | $\mathcal{O}(2Lnrd)$ |
| `CrossSpectra` | $\mathcal{O}(\kappa_1\kappa_2)$ | $\mathcal{O}(nd^2 + D)$ | $\mathcal{O}(nd^2 + D)$ |

Table 1: Complexity comparison. We denote $n$ as sequence length, $L$ is number of layers, and $D = 3Ld^2(2\log d + \log(3L))$ is the 3D FFT cost.

**Algorithm 1** CrossSpectra Forward Pass

1: **Input:** Sparse coefficients $C$ with
2:     $\kappa_1$ non-zeros per layer, $\kappa_2$ layers.
3: $C_{full} \leftarrow$ SparseToDense($C$)
4: $T_{QKV} \leftarrow$ iFFT3D($C_{full}$)
5: **for** $l = 1$ to $L$ **do**
6:     **for** $M \in \{Q, K, V\}$ **do**
7:         $\Delta W_l^M \leftarrow$ Extract($T_{QKV}, M, l$)
8:         $\tilde{W}_l^M \leftarrow W_l^M + \Delta W_l^M$
9:     **end for**
10: **end for**
11: **Return:** $\{\tilde{W}_l^M\}$

**Algorithm 2** CrossSpectra Backward Pass

1: **Input:** Gradients $\{\nabla_{\tilde{W}_l^M}\mathcal{L}\}$.
2: Initialize $\nabla_{T_{QKV}} \in \mathbb{R}^{d \times d \times 3L}$
3: **for** $l = 1$ to $L$ **do**
4:     **for** $M \in \{Q, K, V\}$ **do**
5:         Place $\nabla_{\tilde{W}_l^M}\mathcal{L}$ into
6:           $\nabla_{T_{QKV}}$ at $(M, l)$
7:     **end for**
8: **end for**
9: $\nabla_{C_{full}} \leftarrow$ FFT3D($\nabla_{T_{QKV}}$)
10: $\nabla_C \leftarrow$ SparseSample($\nabla_{C_{full}}, \kappa_1, \kappa_2$)
11: **Return:** $\nabla_C$

Table 2: Performance comparison of LLaMA2 7B with different methods on eight commonsense reasoning datasets. The symbol † indicates that the results are taken from [Wang et al., 2024a, Zhong et al., 2024, Si et al., 2024].

| Method | # Params(%) | BoolQ | PIQA | SIQA | HellaSwag | WinoGrande | ARC-e | ARC-c | OBQA | Average |
|--------|-------------|-------|------|------|-----------|------------|-------|-------|------|---------|
| ChatGPT † | / | 73.10 | 85.40 | 68.50 | 78.50 | 66.10 | 89.80 | 79.90 | 74.80 | 77.01 |
| LoRA† | 0.84 | 69.80 | 79.90 | 79.50 | 83.60 | 82.60 | 79.80 | 64.70 | 81.00 | 77.61 |
| DoRA† | 0.84 | 71.80 | 83.10 | 79.90 | 89.10 | 83.00 | 84.50 | 71.00 | 81.20 | 80.45 |
| PiSSA† | 0.84 | 67.60 | 78.10 | 78.40 | 76.60 | 78.00 | 75.80 | 60.20 | 75.60 | 73.78 |
| MiLoRA† | 0.84 | 67.60 | 83.80 | 80.10 | 88.20 | 82.00 | 82.80 | 68.80 | 80.60 | 79.24 |
| LoRA-Dash† | 0.84 | 71.00 | 75.70 | 79.30 | 91.10 | 78.60 | 84.20 | 69.80 | 78.80 | 78.56 |
| NEAT† | 0.84 | 71.70 | 83.90 | 80.20 | 88.90 | 84.30 | 86.30 | 71.40 | 83.00 | 81.21 |
| KaSA† | 0.84 | 73.60 | **84.40** | 80.20 | **91.50** | 84.50 | 84.70 | 72.10 | 81.20 | 81.53 |
| MoLoRA | 0.96 | 73.15 | 83.68 | 80.09 | 74.57 | 85.95 | 87.33 | 72.53 | 86.20 | 80.43 |
| HydraLoRA | 0.84 | 72.78 | 84.06 | 79.68 | 80.34 | 86.66 | 87.12 | 72.35 | 86.00 | 81.12 |
| CrossSpectra | 0.02 | **73.69** | 83.95 | **80.34** | 88.42 | **87.24** | **87.75** | **76.88** | **87.21** | **82.73** |

The memory efficiency of CrossSpectra stems from sparse spectral parameterization. For typical transformer configurations with $L = 24$ layers and $d = 768$, full fine-tuning requires 42 MB, while LoRA with rank $r = 8$ needs 2.2 MB. CrossSpectra with $\kappa_1 = 1024$ (sparse within layers) and $\kappa_2 = 8$ (cross-layer truncation) requires only 8 KB—achieving 275× reduction compared to LoRA and 5,250× reduction compared to full fine-tuning.

Computationally, CrossSpectra's forward pass consists of a single 3D iFFT operation with complexity $\mathcal{O}(3Ld^2(2\log d + \log(3L)))$, followed by standard matrix multiplications. For sequences of length $n$, the dominant cost remains the attention mechanism's $\mathcal{O}(n^2 d)$ complexity, making CrossSpectra's FFT overhead negligible in practice. The backward pass similarly requires a 3D FFT for gradient computation with the same complexity. Modern FFT implementations on GPU further reduce this overhead through optimized memory access patterns and parallelization.

## 5 Experiments

**Baselines.** To validate the effectiveness of CrossSpectra, we compare against three categories of baselines: 1) **Full FT**: Full fine-tuning of all parameters; 2) **Single-LoRA variants**: LoRA [Hu et al., 2021c], DoRA [Liu et al., 2024], PiSSA [Meng et al., 2024], MiLoRA [Wang et al., 2024b], rsLoRA [Kalajdzievski, 2023], LoRA-Dash [Si et al., 2024], NEAT [Zhong et al., 2024], and KaSA [Wang et al., 2024a]; 3) **LoRA MoE methods**: MoLoRA [Zadouri et al., 2024], AdaMoLE [Liu and Luo, 2024], and HydraLoRA [Tian et al., 2024]. These baselines represent the current state of the art in parameter-efficient fine-tuning, both for standard and mixture-of-experts variants.

**Benchmarks.** To demonstrate the cross-modal effectiveness of our spectral approach, we evaluate CrossSpectra on four diverse tasks. **Image Classification (IC):** We fine-tune CLIP ViT-B/32 [Radford et al., 2021] on 7 standard image datasets including StanfordCars, DTD, EuroSAT, GTSRB, RESISC45, SUN397, and SVHN [Ilharco et al., 2023]. This evaluates CrossSpectra's ability

Table 3: Performance comparison of RoBERTa-large with different methods on 7 GLUE tasks. Total rank is set to 32.

| Method | # Params (%) | CoLA | SST-2 | MRPC | QQP | MNLI | QNLI | RTE | Average |
|---|---|---|---|---|---|---|---|---|---|
| **Full FT** | 100 | 84.27 | 95.98 | 85.29 | 91.58 | 89.83 | 94.49 | 84.84 | 89.47 |
| **LoRA** | 4.00 | 83.41 | 95.64 | 83.33 | 90.06 | 89.00 | 93.28 | 84.47 | 88.46 |
| **DoRA** | 4.00 | 85.33 | 95.99 | 84.07 | 91.24 | 89.52 | 93.54 | 84.48 | 89.17 |
| **PiSSA** | 4.00 | 69.12 | 95.98 | 82.84 | 91.24 | 88.94 | 93.59 | 73.29 | 85.00 |
| **MiLoRA** | 4.00 | 84.65 | 96.10 | 86.02 | 91.33 | 89.51 | 94.12 | 84.83 | 89.51 |
| **rsLoRA** | 4.00 | 83.51 | 95.98 | 86.02 | 90.75 | 88.97 | 93.84 | 84.12 | 89.03 |
| **MoLoRA** | 4.50 | 83.94 | 96.10 | 87.75 | 91.45 | 89.36 | 93.90 | 84.11 | 89.52 |
| **AdaMoLE** | 4.56 | 83.99 | 95.76 | **86.03** | **91.48** | 89.21 | 93.64 | 83.75 | 89.12 |
| **HydraLoRA** | 2.75 | 83.89 | 95.52 | 85.04 | 91.02 | 89.34 | 93.87 | 81.22 | 88.56 |
| CrossSpectra | 0.01 | **86.86** | **96.21** | 84.55 | 91.40 | **89.55** | 94.19 | 85.56 | **89.76** |

to capture visual adaptation patterns. **Natural Language Understanding (NLU):** We fine-tune RoBERTa-large [Liu, 2019] on the GLUE benchmark [Raffel et al., 2020b], which comprises diverse language tasks including grammatical acceptance (CoLA), sentiment analysis (SST-2), paraphrase detection (MRPC, QQP), and natural language inference (MNLI, QNLI, RTE). **Commonsense Reasoning (CR):** We use LLaMA2-7B [Touvron et al., 2023] on 8 reasoning benchmarks: BoolQ, PIQA, SIQA, HellaSwag, WinoGrande, ARC-e, ARC-c, and OBQA. Following Hu et al. [2023], we combine training datasets from all tasks and evaluate on each test set separately. **Arithmetic Reasoning (AR):** Using LLaMA2-7B, we evaluate mathematical reasoning capabilities on GSM8K [Cobbe et al., 2021], MAWPS [Koncel-Kedziorski et al., 2016], SVAMP [Patel et al., 2021], and AQuA [Ling et al., 2017] benchmarks. The training data combines these sources with step-by-step rationales. These diverse tasks let us verify that the cross-layer spectral structure we exploit exists across various model architectures and domains.

**Implementation Details.** For CrossSpectra, we adapt all query, key, and value projection matrices in transformer attention blocks, except in image classification tasks where we only adapt query and key matrices following standard practice. For frequency sparsity, we set the number of non-zero coefficients $|\Omega| = 3000$ (corresponding to approximately $k_1 = 1000$ samples per layer slice and $k_2 = 3$ frequencies in the layer dimension). This represents just 0.1-0.5% of the full parameter space depending on model size. For baseline comparisons, we use LoRA with rank $r = 16$ and $r = 32$. All models are trained using Adam optimizer [Kingma and Ba, 2014] with batch size 64 and cosine learning rate scheduling. For image classification, we use separate learning rates: $1e^{-3}$ for the classification layer and $1e^{-5}$ for adaptation parameters.

## 5.1 Main Results

Table 4: We evaluate CLIP ViT-B/32 with full fine-tuning and LoRA variants with total rank 8 across StanfordCars, DTD, EuroSAT, GTSRB, RESISC45, SUN397, and SVHN datasets. **Bold** indicates the highest results.

| Method | # Params (%) | Cars | DTD | EuroSAT | GTSRB | RESISC45 | SUN397 | SVHN | Average |
|---|---|---|---|---|---|---|---|---|---|
| **Full FT** | 100 | 60.33 | 73.88 | 98.96 | 98.30 | 93.65 | 53.84 | 96.78 | 82.25 |
| **LoRA** | 1.49 | 41.02 | 70.15 | 98.66 | 96.51 | 90.38 | 47.51 | 95.39 | 77.09 |
| **LoRA (rank16)** | 2.99 | 46.51 | 72.07 | 98.74 | 98.04 | 92.08 | 51.63 | 96.00 | 79.30 |
| **LoRA (rank32)** | 5.98 | 50.13 | 72.87 | 98.88 | 98.13 | 92.87 | 53.65 | 96.55 | 80.44 |
| **DoRA** | 1.49 | 40.75 | 71.91 | **98.89** | 97.71 | 90.19 | 47.54 | 95.46 | 77.49 |
| **PiSSA** | 1.49 | 40.41 | 69.62 | 98.48 | 95.84 | 90.58 | 47.21 | 95.84 | 76.85 |
| **MiLoRA** | 1.49 | 39.77 | 70.48 | 98.19 | 97.52 | 89.92 | 45.38 | 95.49 | 76.68 |
| **MoLoRA** | 2.24 | 50.83 | 73.51 | 98.63 | 97.72 | 92.58 | 52.55 | 96.00 | 80.26 |
| **AdaMoLE** | 2.33 | 49.47 | 71.65 | 98.52 | 97.73 | 91.95 | 52.29 | 95.82 | 79.63 |
| **HydraLoRA** | 1.58 | 48.42 | 72.18 | 98.40 | 97.28 | 92.93 | 51.80 | 96.06 | 79.58 |
| CrossSpectra | 0.03 | **53.50** | **75.32** | 98.82 | **98.17** | 93.46 | **54.53** | **96.62** | **81.49** |

The experimental results across multiple modalities in Tables 2,3,4 and 6 demonstrate that CrossSpectra's superior effectiveness in parameter-efficient fine-tuning. On commonsense reasoning tasks with LLaMA2-7B, CrossSpectra achieves 82.73% average accuracy, outperforming methods

Table 5: Performance comparison under different frequency sparsity levels. `CrossSpectra` achieves optimal performance with only 3.2‰ of the frequency space.

| | CSR Task | | | | IC Task | | | |
|---|---|---|---|---|---|---|---|---|
| # Freq. $|\Omega|$ | 10000 | 30000 | 60000 | 120000 | 1000 | 3000 | 6000 | 12000 |
| Sparsity $|\Omega|/(d^2)$ | 0.9‰ | 1.8‰ | 3.6‰ | 7.2‰ | 2.5‰ | 5.0‰ | 10.1‰ | 20.2‰ |
| `CrossSpectra` | **82.15** | **82.73** | **82.33** | **82.65** | **79.18** | **80.34** | **81.49** | **81.32** |

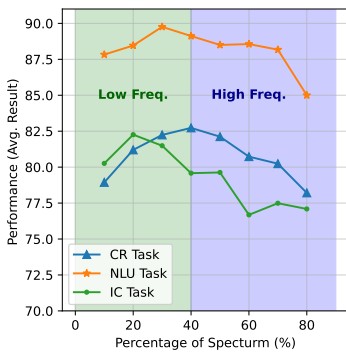

Table 6: Accuracy comparison of various LLMs using PEFT methods on arithmetic reasoning tasks. Results marked with an asterisk (∗) are sourced from Hu et al. [Hu et al., 2023]. (†) denotes our reproduced results on LoRA.

| Model | Methods | MAWPS | SVAMP | GSM8K | AQuA | Avg. |
|---|---|---|---|---|---|---|
| | Base | 51.7 | 32.4 | 15.7 | 16.9 | 24.8 |
| | LoRA* | 79.0 | 52.1 | 37.5 | 18.9 | 44.6 |
| | **DoRA** | 79.2 | 52.0 | 37.3 | 17.4 | 43.1 |
| LLaMA2-7B | **PiSSA** | 78.6 | 52.5 | 38.0 | 17.7 | 44.7 |
| | **MiLoRA** | 79.0 | 52.1 | 37.5 | 18.9 | 44.6 |
| | CrossSpectra | 81.4 | 51.8 | 38.2 | 18.1 | 43.7 |

Figure 3: Left: Performance *vs.* spectrum coverage across modalities. Task performance rapidly increases with low frequencies (0-20%) and shows diminishing returns beyond 60%, validating our theoretical prediction of low-frequency dominance.

such as KaSA (81.53%) and NEAT (81.21%). For NLU tasks using RoBERTa-large, it scores 89.76% across GLUE benchmarks, excelling particularly on complex tasks such as CoLA (86.86%). The cross-modality effectiveness is further validated in vision tasks with CLIP ViT-B/32, CrossSpectra reaches 81.49% average accuracy, surpassing even high-rank LoRA variants (80.44%). For arithmetic reasoning, it outperforms standard approaches on benchmarks such as GSM8K and MAWPS. These consistent results across diverse tasks validate our theoretical framework that exploiting cross-layer smoothness via tensor-based Fourier parameterization enables substantial parameter reduction while maintaining or improving performance, confirming that weight adaptations exhibit the spectral bias predicted by our gradient analysis.

## 5.2 Analysis of Spectral Properties in Weight Adaptations.

The experimental analysis of Figure 1 and Figure 3 provide compelling evidence for our theoretical claims about the spectral properties of weight adaptations in neural networks with skip connections.

**Low-Frequency Dominance in Adaptation Energy.** Figure 1 illustrates the distribution of energy across different frequency components in weight of LLaMA2-7B (we use value projection in the attention layer for illustration). The analysis reveals that a significant portion of the total adaptation energy (69.7%) is concentrated in the low-frequency region of the spectrum, while only 30.3% is distributed across high-frequency components. This striking imbalance confirms weight adaptations should inherit the spectral properties with energy concentrated in low-frequency components.

**Performance Correlation with Spectral Components.** Figure 3 demonstrates how model performance across different tasks (Commonsense Reasoning, Natural Language Understanding, and Image Classification) varies as we progressively increase the sample portions of the frequency spectrum. (*i.e.*, the x-axis indicate we limit the selected frequency basis reside in the first $X\%$ part of spectrum). It demonstrates that across all three modalities (Commonsense Reasoning, Natural Language Understanding, and Image Classification), performance increases rapidly with just the initial low-frequency components ($0-20\%$ of the spectrum), stabilizes in the middle range ($20-60\%$), and shows diminishing returns in the high-frequency range (beyond $60\%$). The consistent pattern across diverse modalities confirms that the smoothness induced by skip connections creates adaptation patterns that are efficiently representable in the low frequency domain along the layer dimension, allowing our approach to achieve strong performance while using dramatically fewer parameters.

Table 7: **Spectral energy *vs.* model scale.** Low-frequency dominance persists across sizes.

| Model | Layers | Params | Low-Freq (%) | High-Freq (%) |
|---|---|---|---|---|
| LLaMA2-13B | 40 | 13B | 61.2 | 38.8 |
| LLaMA2-7B | 32 | 7B | 69.7 | 30.3 |
| RoBERTa-L | 24 | 355M | 68.4 | 31.6 |
| ViT-B/32 | 12 | 86M | 70.9 | 29.1 |

Table 8: **LLaMA2-13B commonsense results.** `CrossSpectra` surpasses LoRA with far fewer parameters.

| Method | BoolQ | PIQA | SIQA | HellaSwag | Avg. |
|---|---|---|---|---|---|
| LoRA | 75.2 | 83.4 | 82.1 | 89.8 | 82.62 |
| CrossSpectra | **76.1** | 83.2 | **82.7** | **92.1** | **83.52** |

Table 9: **Computational efficiency.** Sub-linear time scaling with model size.

| Model | Layers | Time/Epoch (s) | Params(%) | Avg. |
|---|---|---|---|---|
| LLaMA2-13B (CrossSpectra) | 40 | 1.82 | 0.02 | 83.52 |
| LLaMA2-7B (CrossSpectra) | 32 | 0.79 | 0.02 | 82.73 |
| LLaMA2-7B (LoRA) | 32 | 0.61 | 0.84 | 77.61 |

Table 10: **FFT overhead.** 3D iFFT cost is negligible *vs.* attention.

| Tensor Size | FFT Time (s) |
|---|---|
| $(4096, 4096, 3 \times 10)$ | 0.05 |
| $(4096, 4096, 3 \times 20)$ | 0.12 |
| $(4096, 4096, 3 \times 30)$ | 0.28 |
| $(4096, 4096, 3 \times 40)$ | 0.41 |

**Sparsity along the Weight Dimension.** Table 6 presents a comprehensive comparison of various PEFT methods' performance specifically on arithmetic reasoning tasks across different LLM architectures ( LLaMA2-7B). The evaluation spans four key mathematical benchmarks: MAWPS (math word problems), SVAMP (simple variations on arithmetic problems), GSM8K (grade school math), and AQuA (arithmetic questions and answers). The results demonstrate that `CrossSpectra` achieves comparable or superior performance to standard LoRA and DoRA methods in solving these complex mathematical problems.

**Impact of Frequency Sparsity.** Table 5 demonstrates how `CrossSpectra`'s performance varies with different levels of frequency sparsity across modalities. For commonsense reasoning (CSR) tasks, we tested sparsity levels from 0.9‰ to 7.2‰ of the full frequency space (10,000 to 120,000 non-zero frequencies). Similarly, for image classification (IC) tasks, we explored sparsity levels from 2.5‰ to 20.2‰ (1,000 to 12,000 frequencies). Remarkably, `CrossSpectra` achieves optimal performance with extremely sparse parameterization—only 1.8‰ (30,000 frequencies) for CSR tasks and 5.0‰ (3,000 frequencies) for IC tasks. This confirms our theoretical prediction that weight adaptations naturally concentrate in a small subset of frequency components. The consistent performance across wide sparsity ranges further validates that our approach captures the essential adaptation information while eliminating redundancy. Even at the sparsest setting (0.9‰), `CrossSpectra` outperforms traditional methods that use orders of magnitude more parameters, demonstrating the practical impact of our theoretical insights about spectral concentration in transformer adaptations.

**Scaling to Larger Models.** We examine whether the low-frequency concentration holds as model size increases. Table 7 shows that the low-frequency share of adaptation energy remains high (61–71%) from ViT-B/32 (86M) up to LLaMA2-13B (13B), supporting our scale-agnostic hypothesis. We further fine-tune LLaMA2-13B on commonsense reasoning. `CrossSpectra` surpasses LoRA with far fewer parameters (Table 8), indicating that our cross-layer spectral parameterization scales favorably.

**Computational Efficiency.** Training time scales sub-linearly with model size due to sparse spectral coefficients (Table 9). LoRA is faster per epoch but yields lower accuracy, illustrating an efficiency–accuracy trade-off. Using `torch.fft.ifftn`, 3D FFT runtime grows as $\mathcal{O}(N \log N)$ and remains negligible relative to attention (Table 10).

## 6 Conclusions

This work connects neural network theory with efficient adaptation methods, revealing how architectural properties determine optimal weight adaptation parameterization. `CrossSpectra` demonstrates that skip connections create structured adaptation patterns across layers—an insight previously overlooked. Future research could explore dynamic spectral bases, adaptation for emerging architectures, or information-theoretic compression limits. Our findings suggest the most efficient neural network parameterizations align with their intrinsic dynamics rather than treating parameters independently. As foundation models grow, leveraging these inherent structures will be essential for making specialized adaptation accessible.

## Acknowledgments

The research is supported, in part, by the Ministry of Education, Singapore, under its Academic Research Fund Tier 1 (RG101/24); the RIE2025 Industry Alignment Fund – Industry Collaboration Projects (IAF-ICP) (Award I2301E0026), administered by A*STAR, as well as supported by Alibaba Group and NTU Singapore through Alibaba-NTU Global e-Sustainability CorpLab (ANGEL). Piotr Koniusz and Hao Zhu are supported by CSIRO's Science Digital.

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

# A  Proofs

Our analysis builds on standard stability and smoothness properties of residual architectures.

**Assumption A.1** (Residual stability and ODE scaling). *Each residual block $R_l(H_{l-1}; W_l)$ is Lipschitz continuous with $\|\partial R_l/\partial H_{l-1}\|_2 \leq c_H$ and $\|\partial R_l/\partial W_l^M\|_2 \leq c_M$, where $c_H, c_M$ are depth-independent constants. We assume residual–ODE scaling $H_l = H_{l-1} + \frac{1}{L} R_l(H_{l-1}; W_l)$, and bounded loss gradients $\|\partial \mathcal{L}/\partial H_L\|_F \leq G$ (Xiong et al., 2020, Liu et al., 2020). Layer normalization and residual scaling guarantee such bounded Jacobians.*

**Assumption A.2** (Spatial regularity and weak separability). *Within-layer adaptation maps possess bounded Sobolev norms, implying decay of high-frequency spectral components in each spatial dimension. Such spectral bias is well documented in neural networks (Rahaman et al., 2019). Moreover, the 3-D adaptation tensor is approximately separable across spatial and layer dimensions, as suggested by empirical analyses of attention weight spectra.*

These assumptions are mild and supported by empirical evidence in prior work. They ensure that the subsequent proofs yield depth-normalized gradient smoothness and power-law spectral decay.

## A.1  Proof of Theorem 3.2

*Proof.* **Step 1: Residual formulation.** We express each transformer block in residual–ODE form $H_l = H_{l-1} + \frac{1}{L} R_l(H_{l-1}; W_l)$, where $R_l$ denotes the combined attention and feed-forward submodules. Let $J_j = \partial R_j/\partial H_{j-1}$ and $K_l = \partial R_l/\partial W_l^M$, with $\|J_j\|_2 \leq c_H$ and $\|K_l\|_2 \leq c_M$.

**Step 2: Gradient decomposition.** By the chain rule,

$$\nabla_{W_l^M}\mathcal{L} = \frac{\partial \mathcal{L}}{\partial H_L}\frac{\partial H_L}{\partial W_l^M}, \qquad \frac{\partial H_L}{\partial W_l^M} = \frac{1}{L}\sum_{t=l}^{L}\Phi_{t+1\leftarrow l}K_l,$$

where the propagation operator $\Phi_{t+1\leftarrow l} = \prod_{j=l+1}^{t}(I + \frac{1}{L}J_j)$ satisfies $\|\Phi_{t+1\leftarrow l}\|_2 \leq e^{c_H(t-l)/L}$.

**Step 3: Bounding cross-layer differences.** Successive gradients differ by

$$\frac{\partial H_L}{\partial W_{l+1}^M} - \frac{\partial H_L}{\partial W_l^M} = \frac{1}{L}\sum_{t=l+1}^{L}(\Phi_{t+1\leftarrow l+1} - \Phi_{t+1\leftarrow l})K_l.$$

Using the series expansion $\Phi_{t+1\leftarrow l+1} - \Phi_{t+1\leftarrow l} = O(\frac{1}{L})$ and $\|\Phi_{t+1\leftarrow l}\|_2 \leq e^{c_H(t-l)/L}$ gives

$$\left\|\frac{\partial H_L}{\partial W_{l+1}^M} - \frac{\partial H_L}{\partial W_l^M}\right\|_2 \leq \frac{(e^{c_H}-1)c_M}{L}.$$

Multiplying by $\|\partial \mathcal{L}/\partial H_L\|_F \leq G$ yields

$$\|G_{l+1}^M - G_l^M\|_F \leq \frac{C_M}{L}, \qquad C_M = G\,c_M(e^{c_H}-1).$$

$\square$

## A.2  Proof of Theorem 3.3

*Proof.* **Step 1: Constructing the adaptation tensor.** Let $\Delta W_l^M$ denote the accumulated weight updates, and form $A^M \in \mathbb{R}^{d\times d\times L}$ with $A_{:,:,l}^M = \Delta W_l^M$. Define its 3D discrete Fourier transform:

$$\widehat{A}^M(n_1, n_2, n_3) = \sum_{i_1,i_2,l} A_{i_1,i_2,l}^M\, e^{-2\pi i(n_1 i_1/d + n_2 i_2/d + n_3 l/L)}.$$

**Step 2: Smoothness implies spectral decay.** From Theorem 3.2,

$$\|\Delta W_{l+1}^M - \Delta W_l^M\|_F \leq \frac{C_1}{L}.$$

A bounded first finite difference yields Fourier magnitude

$$\|\widehat{\boldsymbol{A}}^M_{:,:,n_3}\|_F \leq \frac{\tilde{C}_1}{1 + n_3}.$$

If a stronger second-difference bound $\|\Delta \boldsymbol{W}^M_{l+1} - 2\Delta \boldsymbol{W}^M_l + \Delta \boldsymbol{W}^M_{l-1}\|_F \leq C_2/L^2$ holds, then

$$\|\widehat{\boldsymbol{A}}^M_{:,:,n_3}\|_F \leq \frac{\tilde{C}_2}{(1 + n_3)^2}.$$

This confirms a power-law decay in the cross-layer dimension.

**Step 3: Combined spatial–layer spectral structure.** Empirically, neural networks exhibit spectral bias toward low within-layer frequencies [Rahaman et al., 2019], so that

$$|\widehat{\boldsymbol{A}}^M(n_1, n_2, n_3)| \leq \frac{C_M}{(1 + n_1)^{\beta_1} (1 + n_2)^{\beta_2} (1 + n_3)^{\beta_3}},$$

where $\beta_3 > \beta_1, \beta_2$ due to stronger smoothness along depth. This establishes the claimed dimension-specific spectral decay.

$\square$

# B   Limitations and Future Work

While `CrossSpectra` demonstrates significant parameter efficiency across diverse tasks, there are several limitations worth noting. First, our current approach focuses exclusively on adapting attention weights $(\boldsymbol{Q}, \boldsymbol{K}, \boldsymbol{V})$, potentially missing optimization opportunities in other components like feed-forward networks. This design choice has the benefit of reducing overfitting risk by targeting the most information-dense parameters identified by our gradient analysis, but extending the spectral approach to other transformer components in a principled way could yield further improvements. Second, the computational overhead of 3D FFT operations, while well-optimized on modern hardware, might become a bottleneck for extremely large models (trillions of parameters) or resource-constrained deployment environments. Third, our method requires a careful selection of frequency sparsity patterns—currently uniform sampling within layers—which may not be optimal for all model architectures or tasks. A promising direction for future work is to develop adaptive frequency sampling strategies that automatically identify the most important spectral components for a specific task. Additionally, exploring theoretical connections between `CrossSpectra` and other parameter-efficient methods like prompt tuning could lead to hybrid approaches that combine their complementary strengths. Finally, while we observed consistent cross-layer spectral patterns across model sizes up to 7B parameters, verifying that these properties scale efficiently to models with hundreds of billions of parameters remains an important direction for future research.

