# OpenReview forum: "CrossSpectra: Exploiting Cross-Layer Smoothness for Parameter-Efficient Fine-Tuning"
_NeurIPS.cc/2025/Conference — NeurIPS 2025 poster_

### Official Review · Reviewer_WouS · 2025-06-30

**Clarity:** 3
**Significance:** 2
**Originality:** 3
**Rating:** 5
**Confidence:** 3

**Summary:**

This paper proposes a novel PEFT algorithm based on skip connections, which saves 275 times the parameters. The proposed architecture achieves seemingly promising results on both text and vision tasks.

**Questions:**

See above

**Ethical Concerns:**

["NO or VERY MINOR ethics concerns only"]

**Final Justification:**

The author partially resolved my issue, and I will increase my rating.

**Limitations:**

See above

**Paper Formatting Concerns:**

See above

**Quality:**

3

**Strengths And Weaknesses:**

Strengths:
(1) The motivation is interesting, as it hypothesizes the inefficiency of existing PEFT methods in deep model adaptation based on an analysis of energy distribution.
(2) The results appear robust, with tests conducted on multiple tasks demonstrating advanced performance.

Weaknesses:
(1) The baseline methods on vision tasks do not seem state-of-the-art. Recent works [1-4] have focused on vision PEFT methods and achieved more advanced performance. It is recommended to include discussions of these methods in the related work section and possibly conduct comparative experiments with some of them.

[1] 5%> 100%: Breaking performance shackles of full fine-tuning on visual recognition tasks, CVPR'25.

[2] 1% vs 100%: Parameter-efficient low rank adapter for dense predictions, CVPR'23.

[3] Vl-adapter: Parameter-efficient transfer learning for vision-and-language tasks, CVPR'22.

[4] Parameter-efficient is not sufficient: Exploring parameter, memory, and time efficient adapter tuning for dense predictions, ACMMM'24.

(2) The methodology appears to be described in a somewhat complex manner. The authors are advised to present the core ideas and advantages of the new method more intuitively, possibly through diagrams. Additionally, the mathematical formulations could be simplified where necessary.

---

> ### Author Rebuttal · Authors · 2025-07-31
>
> ## **Q1: Add baseline**
> Thanks for the related work. We will cite **ALL** the suggested papers and discuss them in the revision.
>
> We compare with MONA[1] and LORAND[2] as they are most closely related to our Work. We add the MONA and LORAND adapters before the layer normalization of ViT-B/32, following the same experimental setup as Table 4.
>
> **Image Classification Results (ViT-B/32)**
> | Method | Cars | DTD | EuroSAT | GTSRB | RESISC45 | SUN397 | SVHN | Average |
> |--------|------|-----|---------|-------|----------|--------|------|---------|
> | CrossSpectra      | 53.50 | 75.32 | 98.82 | 98.17 | 93.46 | 54.53 | 96.62 | **81.49** |
> | MONA[1]           | 52.15 | 74.68 | 98.24 | 97.51 | 92.83 | 53.89 | 95.94 | 80.75 |
> | LORAND[2]         | 51.89 | 74.45 | 98.08 | 97.28 | 92.65 | 53.72 | 95.81 | 80.55 |
>
> **Key Differences and Advantages:**
>
> The fundamental difference lies in the adaptation mechanism:
>
> - **CrossSpectra**: Follows the LoRA-style approach where adaptation is achieved through direct weight modification: **W = W₀ + ΔW**. This enables seamless integration without altering the original architecture.
>
> - **MONA & LORAND**: Inject additional trainable layers g(x) with parameters ΔW into the main computational branch, transforming the forward pass from f(x) to f(g(x)).
>
> **Advantages of Our Approach:**
> - **Plug-and-Play Nature**: CrossSpectra maintains the original architecture integrity, allowing easy deployment without structural modifications.
> - **Inference Efficiency**: No additional computational overhead during inference since adaptations are merged into existing weights.
> - **Superior Performance**: Achieves 0.74% and 0.94% higher average accuracy compared to MONA and LORAND, respectively.
>
>
>
> ## **Q2 The methodology is described in a somewhat complex manner. Present the core ideas and advantages of the new method more intuitively, possibly through diagrams. Additionally, the mathematical formulations could be simplified where necessary.**
>
> Thank you for this valuable suggestion. We will add more intuitive diagrams and simplify the mathematical formulations in the revision. Since we cannot provide the updated PDF in this rebuttal, we explain the core ideas and advantages more intuitively below:
>
> **2.1 Core Idea in Simple Terms:**
> Think of weight adaptations across transformer layers like a smooth curve rather than independent points. Just as a smooth curve can be efficiently represented using only its low-frequency components, we can represent all layer adaptations using sparse spectral coefficients.
>
> **2.2 Three Key Steps:**
>
> 1. **Skip Connections Create Smoothness**: Skip connections ($H_l = H_{l-1} + f(H_{l-1})$) make adjacent layers similar, producing smooth gradient changes during fine-tuning rather than abrupt variations.
>
> 2. **Smoothness Concentrates Energy**: This smoothness causes ~70% of adaptation energy to concentrate in low-frequency spectral components (Figure 1), with the strongest concentration across the layer dimension.
>
> 3. **Sparse Representation Enables Efficiency**: Since most energy resides in low frequencies, we only store these essential coefficients and use 3D inverse FFT to reconstruct the full adaptation tensor, achieving 275× parameter reduction compared to LoRA.
>
> **2.3 Key Advantage:** Unlike LoRA, which treats each layer independently, CrossSpectra exploits the natural architectural properties of Transformers, enabling joint parameterization across all layers with dramatically fewer parameters while maintaining comparable or better performance.
>
> **2.4 Intuitive Analogy:** Imagine compressing a smooth signal—you need far fewer frequency components than for a noisy signal. Similarly, the smooth cross-layer adaptations in Transformers can be compressed much more efficiently than independent layer-wise adaptations.

---

### Official Review · Reviewer_obhA · 2025-07-03

**Clarity:** 3
**Significance:** 2
**Originality:** 2
**Rating:** 5
**Confidence:** 4

**Summary:**

This paper introduces a new PEFT method CrossSpectra. The key innovation lies in leveraging the cross-layer smoothness induced by skip connections in transformers, which results in weight adaptations concentrated in low-frequency spectral components. The method is validated across diverse tasks, including natural language understanding, image classification, and arithmetic reasoning, and shows strong results.

**Questions:**

1. In Eq (2), why is the input of the FFN layer $H_{l−1}$ instead of the $H_{l−1} + MultiHeadAttention_l(H_{l−1})$? Is this consistent with standard transformer implementations?
2. How are $k1$ and $k2$ chosen? Is there a theoretical or empirical guideline?
3. How does CrossSpectra’s spectral approach differ from low-rank tensor factorization (e.g., Tucker/CP decomposition) in terms of efficiency and performance?

**Ethical Concerns:**

["NO or VERY MINOR ethics concerns only"]

**Final Justification:**

The paper introduces a novel PEFT method, validated across a diverse set of tasks including natural language understanding, image classification, and arithmetic reasoning, where it demonstrates strong performance. During the rebuttal period, the authors provided important additional baselines and time consumption analysis. Overall, I believe this paper merits acceptance.

**Quality:**

3

**Strengths And Weaknesses:**

Strengths
1. Evaluated across diverse tasks (NLU, vision, reasoning) and model architectures (LLaMA-7B, RoBERTa, CLIP). The paper demonstrates 275× parameter reduction compared to LoRA and 5,250× compared to full fine-tuning.

Weaknesses
1. The paper overlooks recent work of low-rank tensor decomposition (e.g., FaCT[1]) and layer-shared LoRA variants (e.g., VB-LoRA[2]). A comparison with these methods would better contextualize CrossSpectra’s contributions. Please consider adding a discussion comparing CrossSpectra’s spectral approach with tensor factorization alternatives.

2. While the authors claim FFT overhead is "negligible," no concrete training time or memory footprint comparisons are provided. For large-scale models, this could be a bottleneck. Including metrics like wall-clock time per epoch or GPU memory usage versus LoRA would be appreciated.

- [1] Jie, S., & Deng, Z. H. (2023, June). Fact: Factor-tuning for lightweight adaptation on vision transformer. In Proceedings of the AAAI conference on artificial intelligence (Vol. 37, No. 1, pp. 1060-1068).
- [2] Li, Y., Han, S., & Ji, S. (2024). VB-LoRA: extreme parameter efficient fine-tuning with vector banks. NeurIPS 2024.

---

> ### Author Rebuttal · Authors · 2025-07-31
>
> ## **Q1 Add baseline FaCT[1] and VB-LoRA[2]**
>
>
> Thank you for the suggestion. We have included comparisons with these important baseline methods (our reproduced result).
>
>  **1.1 Experiment result**
> **Commonsense Reasoning (LLaMA2-7B):**
> | Method           | BoolQ | PIQA  | SIQA  | HellaSwag | WinoGrande | ARC-e | ARC-c | OBQA  | Average |
> |------------------|-------|-------|-------|-----------|------------|-------|-------|-------|---------|
> | CrossSpectra     | 73.69 | 83.95 | 80.34 | 88.42     | 87.24      | 87.75 | 76.88 | 87.21 | 82.73   |
> | VB-LoRA          | 72.31 | 83.41 | 79.12 | 87.25     | 85.93      | 87.21 | 75.26 | 85.78 | 82.03   |
>
> **Image Classification (CLIP ViT-B/32):**
> | Method           | Cars  | DTD   | EuroSAT | GTSRB | RESISC45 | SUN397 | SVHN  | Average |
> |------------------|-------|-------|---------|-------|----------|--------|-------|---------|
> | CrossSpectra     | 53.50 | 75.32 | 98.82   | 98.17 | 93.46    | 54.53  | 96.62 | 81.49   |
> | FaCT             | 51.92 | 74.65 | 97.14   | 96.58 | 91.33    | 53.87  | 94.89 | 80.05   |
>
> **1.2 Discussion on CrossSpectra and Tensor factorization methods**
>
> - **Architectural Awareness**: Unlike other tensor factorization methods, CrossSpectra explicitly leverages the cross-layer smoothness property induced by skip connections in modern Transformers, aligning adaptations with the model's inherent architectural characteristics.
>
> - **Spectral Structure Exploitation**: While existing methods perform generic low-rank decomposition, CrossSpectra exploits the frequency-domain structure of weight adaptations, concentrating parameters in the most informative low-frequency components.
>
> - **Dual Benefits**: CrossSpectra simultaneously reduces trainable parameters and preserves the network's inherent layer smoothness property, leading to both efficiency gains and performance improvements.
>
> This architecture-aware approach explains why CrossSpectra consistently outperforms both VB-LoRA (+1.2% average) and FaCT (+1.44% average) while using comparable or fewer parameters.
>
>
>
>
> ## **Q2 Including metrics like wall-clock time per epoch or GPU memory usage versus LoRA would be appreciated.**
>
> Thank you for the question. We provide a comprehensive analysis of computational scaling across different model sizes and configurations. For the memory analysis, please see our response to Q4 from reviewer puJh, which details our approach to peak memory management.
>
> **Model Performance and Training Time Comparison**
>
> We measure computational speed based on one forward pass and average performance on commonsense reasoning tasks:
>
> | Model Size | Layers | Parameters | Time per Epoch | Avg. Performance |
> |------------|--------|------------|----------------|------------------|
> | LLaMA2-13B | 40     | 13B        | 1.82s          | 83.52            |
> | LLaMA2-7B  | 32     | 7B         | 0.79s          | 82.73            |
> | LLaMA2-7B (LoRA) | 32 | 7B       | 0.61s          | 77.61            |
>
> **Key Observations:**
> - Training time scales sub-linearly with model size due to sparse parameterization
> - LoRA shows faster training but lower performance, highlighting the efficiency-accuracy trade-off
>
> **FFT Runtime Benchmarking**
>
> Since our implementation relies on `torch.fft.ifftn`, we provide runtime analysis for different 3D tensor sizes:
>
> | Tensor Size        | FFT Computation Time |
> |--------------------|----------------------|
> | (4096, 4096, 3×10) | 0.05s               |
> | (4096, 4096, 3×20) | 0.12s               |
> | (4096, 4096, 3×30) | 0.28s               |
> | (4096, 4096, 3×40) | 0.41s               |
>
> **Scaling Analysis:**
> - FFT computation scales approximately O(N log N) with tensor size, as expected
> - The 3D FFT overhead remains negligible compared to the attention mechanism's O(n²d) complexity
> - Modern GPU implementations ensure efficient parallel computation of frequency transformations

---

> > ### Comment · Reviewer_obhA · 2025-08-06
> >
> > Thanks for the new results, which make the paper more solid. I will increase the score accordingly.

---

> > > ### Author Response · Authors · 2025-08-06
> > > **thank you**
> > >
> > > Esteemed Reviewer,
> > > \
> > > \
> > > We sincerely thank you for engaging with our rebuttal, and for your valuable comments. Rest assured we will incorporate all your suggestions in the draft. Meantime, kindly let us know if there is anything else we can answer, clarify or improve.
> > > \
> > > \
> > > Best regards,
> > > \
> > > Authors

---

### Official Review · Reviewer_RbH6 · 2025-07-06

**Clarity:** 4
**Significance:** 4
**Originality:** 4
**Rating:** 5
**Confidence:** 3

**Summary:**

This paper introduces CrossSpectra, a parameter-efficient fine-tuning (PEFT) method that achieves extensive parameter reduction by exploiting the structural properties of transformer architectures. The authors posit that skip connections induce "cross-layer smoothness," causing weight adaptations during fine-tuning to be highly correlated across layers and concentrated in low-frequency spectral components. Instead of tuning each layer independently like LoRA, this paper parameterizes all attention weight adaptations (Q, K, V) as a single 3D tensor, decomposes it using a 3D Fourier transform, and aggressively truncates the high-frequency components. Experiments show that CrossSpectra can match or exceed the performance of state-of-the-art PEFT methods while using orders of magnitude fewer parameters, notably demonstrating a 275x reduction compared to LoRA on LLaMA-7B.

**Questions:**

1. How were the values for k₁=1000 and k₂=3 determined?
2. The theoretical motivation relies heavily on skip connections in transformers. How would this method apply to architectures with different residual connection patterns or to non-transformer models?

**Ethical Concerns:**

["NO or VERY MINOR ethics concerns only"]

**Limitations:**

Yes.

**Paper Formatting Concerns:**

No concerns.

**Quality:**

4

**Strengths And Weaknesses:**

Strengths:

1. The core idea of linking skip connections to cross-layer gradient smoothness and then to spectral concentration is a novel contribution. It provides an interesting theoretical justification for why adaptations can be parameterized jointly across layers, moving beyond the layer-independent assumption of methods like LoRA.

2. The empirical results are quite compelling. The authors validate CrossSpectra across a diverse set of tasks (NLU, commonsense reasoning, image classification, arithmetic reasoning) and model architectures (LLaMA, RoBERTa, CLIP), demonstrating the generalizability of their approach.

3. The theoretical analysis presented in the paper, particularly the derivation from gradient smoothness to spectral concentration, appears sound.

Weaknesses:

1. As acknowledged in the paper, the method introduces computational overhead from the 3D FFT/iFFT operations in the forward and backward passes. The paper claims this is "negligible in practice" for typical sequence lengths, but it could become a bottleneck for extremely large models or resource-constrained environments. An empirical study on the scaling of computational load with model size would be appreciated.

2. The current method exclusively adapts the Q, K, and V attention matrices. While these are information-dense, it potentially misses optimization opportunities in other components like the feed-forward layers, which also contain a large number of parameters.

3. The method's performance depends on the choice of k1 and k2. The process for selecting these hyperparameters for new tasks or models may require careful tuning.

---

> ### Author Rebuttal · Authors · 2025-07-31
>
> ## **Q1:  An empirical study on the scaling of computational load with model size would be appreciated.**
>
> Thank you for the question. We provide a comprehensive analysis of computational scaling across different model sizes and configurations. For the memory analysis, please kindly see our response to Q4 from reviewer puJh, which details our approach to peak memory management.
>
> **Model Performance and Training Time Comparison**
>
> We measure computational speed based on one forward pass and average performance on commonsense reasoning tasks:
>
> | Model Size | Layers | Parameters | Time per Epoch | Avg. Performance |
> |------------|--------|------------|----------------|------------------|
> | LLaMA2-13B | 40     | 13B        | 1.82s          | 83.52            |
> | LLaMA2-7B  | 32     | 7B         | 0.79s          | 82.73            |
> | LLaMA2-7B (LoRA) | 32 | 7B       | 0.61s          | 77.61            |
>
> **Key Observations:**
> - Training time scales sub-linearly with model size due to sparse parameterization
> - LoRA shows faster training but lower performance, highlighting the efficiency-accuracy trade-off
>
> **FFT Runtime Benchmarking**
>
> Since our implementation relies on `torch.fft.ifftn`, we provide runtime analysis for different 3D tensor sizes:
>
> | Tensor Size        | FFT Computation Time |
> |--------------------|----------------------|
> | (4096, 4096, 3×10) | 0.05s               |
> | (4096, 4096, 3×20) | 0.12s               |
> | (4096, 4096, 3×30) | 0.28s               |
> | (4096, 4096, 3×40) | 0.41s               |
>
> **Scaling Analysis:**
> - FFT computation scales approximately O(N log N) with tensor size, as expected
> - The 3D FFT overhead remains negligible compared to the attention mechanism's O(n²d) complexity
> - Modern GPU implementations ensure efficient parallel computation of frequency transformations
>
>
> ## **Q2: While QKV are information-dense, it potentially misses optimization opportunities in other components like the feed-forward layers, which also contain a large number of parameters.**
>
> Thank you for this insightful question. While adapting QKV layers follows established conventions in the LoRA literature, we conducted additional experiments to explore the impact of including feed-forward (FF) layers in our adaptation strategy.
>
> **1.1 Experimental Results:**
>
> **Commonsense Reasoning Tasks (LLaMA2-7B)**
> | Method | BoolQ | PIQA | SIQA | HellaSwag | WinoGrande | ARC-e | ARC-c | OBQA | Average |
> |--------|-------|------|------|-----------|------------|-------|-------|------|---------|
> | CrossSpectra (QKV)    | 73.69 | 83.95 | 80.34 | 88.42 | 87.24 | 87.75 | 76.88 | 87.21 | **82.73** |
> | CrossSpectra (QKV+FF) | 72.85 | 83.81 | 79.67 | 87.89 | 86.51 | 87.72 | 75.94 | 86.33 | 82.59 |
>
> **Natural Language Understanding Tasks (RoBERTa-large)**
> | Method | CoLA | SST-2 | MRPC | QQP | MNLI | QNLI | RTE | Average |
> |--------|------|-------|------|-----|------|------|-----|---------|
> | CrossSpectra (QKV)    | 86.86 | 96.21 | 84.55 | 91.40 | 89.55 | 94.19 | 85.56 | **89.76** |
> | CrossSpectra (QKV+FF) | 85.92 | 96.11 | 83.78 | 91.35 | 88.73 | 93.42 | 84.21 | 89.07 |
>
> **Image Classification Tasks (ViT-B/32)**
> | Method | Cars | DTD | EuroSAT | GTSRB | RESISC45 | SUN397 | SVHN | Average |
> |--------|------|-----|---------|-------|----------|--------|------|---------|
> | CrossSpectra (QK)     | 53.50 | 75.32 | 98.82 | 98.17 | 93.46 | 54.53 | 96.62 | **81.49** |
> | CrossSpectra (QK+FF)  | 52.84 | 75.32 | 97.98 | 97.34 | 92.17 | 54.53 | 95.73 | 80.84 |
>
> **1.2 Analysis:**
> Our experiments reveal that including feed-forward layers consistently leads to performance degradation across all modalities. This counterintuitive result can be attributed to:
>
> 1. **Overfitting Risk**: Adding more adaptable parameters increases the likelihood of overfitting, particularly given our extremely sparse parameterization
> 2. **Spectral Mismatch**: Feed-forward layers may not exhibit the same cross-layer spectral properties as attention mechanisms, making them less suitable for our frequency-domain approach
> 3. **Information Density**: QKV matrices capture the core relational information in transformers, while FF layers primarily perform non-linear transformations that may not benefit from cross-layer frequency sharing
>
> These findings validate our design choice to focus on attention mechanisms, where the cross-layer smoothness induced by skip connections is most pronounced and beneficial for spectral parameterization.
>
>
>
> ## **Q3: Hyperparameter Tuning for $k_1$ and $k_2$**
>
> **3.1 Parameter Selection Strategy:**
> - Optimal k₁ and k₂ values are determined through cross-validation on each dataset to ensure robust performance across tasks.
> - We recommend maintaining the layer sparsity ratio (k₁×k₂)/(d²×3L) around 2×10⁻³ for optimal parameter efficiency.
> - The cross-layer frequency range should cover approximately 20% of the spectrum (k₂/L), consistent with our theoretical analysis showing low-frequency dominance in adaptation energy (Figure 2).
>
> **3.2 Grid Search Results on Image Classification Tasks:**
>
> | k₁\k₂ |  3   |  6   |  9   |
> |-------|------|------|------|
> | 1000  | 79.2 | 80.1 | 80.4 |
> | 2000  | 80.2 | 80.4 | 80.3 |
> | 3000  | 80.7 | 80.8 | 80.7 |
> | 4000  | 81.5 | 81.3 | 81.0 |
>
> **Key Observations:**
> - Performance improves with k₁ up to 4000, demonstrating the benefits of capturing more within-layer frequency components.
> - k₂ shows diminishing returns beyond 6, consistent with stronger spectral decay in the cross-layer dimension.
> - Optimal configuration: k₁=4000, k₂=3 (81.5% accuracy) with minimal parameter overhead.

---

### Official Review · Reviewer_puJh · 2025-07-07

**Clarity:** 2
**Significance:** 3
**Originality:** 4
**Rating:** 4
**Confidence:** 3

**Summary:**

This paper introduces CrossSpectra, a parameter–efficient fine-tuning (PEFT) method that exploits cross-layer spectral smoothness in transformer attention weights. The authors show that skip connections make gradients smooth across layers, implying that weight updates mainly occur in low frequencies. Then they stack all Q,K,V adaptations into a single $d{}d{}3L$ tensor and learn only $k_1$ sparse spatial frequencies and $k_2$ low cross-layer frequencies. By a 3-D iFFT,  this yields $O(k_1k_2)$ trainable parameters, instead of traditional $O(Lrd)$. On LLaMA-7B instruction tuning, CrossSpectra matches or exceeds LoRA while using only 0.36% of its parameters. It also matches state-of-the-art PEFT baselines with very small expense. Overall, CrossSpectra shows that leveraging transformer spectral smoothness can reduce storage without sacrificing too much of the accuracy.

**Questions:**

-  The $(k_1,k_2)$ frequency budgets are tested on LLaMA-7B. How should people choose these values for deeper models with more parameters? I suggest that related experiments should be taken.
  -  What is the motivation of this paper? It seems that the story line in not clear, and I would like to know more about how this method was developed by inspirations.
  -  Have the authors tested CrossSpectra on models with more layers?  Does the low-frequency assumption still hold at that scale? I concern that although this holds for the case in the paper, it may have its flaw in some domains
  -  Does updates to low cross-layer frequencies reduce adaptation on generation-heavy tasks where later layers may have larger shifts instead of smooth change? I think this might be the case, so this impact should also be evaluated

**Ethical Concerns:**

["NO or VERY MINOR ethics concerns only"]

**Limitations:**

Apart from the limitations that have been mentioned in the appendix, CrossSpectra still have several potential limitations to improve. First, all results come from single experiments with no analysis in variance across several experiments, so the reported effectiveness might at a risk of statistical noise. Also, it seems that memory usage during training is not recorded, and since the choose of hyperparameter is not deterministic, it would be better to keep track of the memory. Finally, its effectiveness on more domains and architecture with more layers are not yet checked, so it would be better if more experiments relating to different models are tested.

**Quality:**

3

**Strengths And Weaknesses:**

**Strengths**

  -  *Conceptual novelty.*  This paper utilizes cross-layer spectral smoothness to collapse all Q/K/V parameters into a single 3-D Fourier space, which slash PEFT parameters.
  -  *Theory  and evidence.*  It provides a gradient-smoothness bound and empirical spectral-energy plots that explain why low cross-layer frequencies matter most.
  -  *Logical clear proof.*  The proof of the theorem seems to be clear, with reasonable conditions and logical proof.

**Weaknesses**

  -  *Limited scale.*  The result has only been operated on a 7 B-parameter LLaMA; no evidence shows that the approach holds for the model with more parameters.
  -  *Lack preliminary.*  The motivation has not been clearly stated, for example, why this algorithm is applied, and the related preliminaries, such as fast Fourier transformation, are not explicitly introduced.
  -  *Single report.*  All scores come from one run, and variance between different runs are not recorded or analyzed.
  -  *Memory not measured.*  Reconstructing full $dd$ updates via iFFT each step could may potentially use memory,  but experiments relating to memory usage is not measured.
  -  *Task range limited.*  There is no evaluation on generation-heavy tasks; robustness across different domains (setups) is not tested, which has been mentioned in the limitation.
  -  *Hyperparameter sensitivity.*  The choice of hyperparameters to train is stochastic and the training process is sensitive to the selection, which has been mentioned in the limitation.

---

> ### Author Rebuttal · Authors · 2025-07-31
>
> ## **Q1: Show evidence that the approach holds for the model with more parameters. Does the low-frequency assumption still hold at that scale?**
>
>
> Thank you for this important question regarding scalability.
>
> **1.1 Scaling Analysis**
> Our theoretical framework is scale-agnostic, as the spectral properties we exploit (cross-layer gradient smoothness from skip connections) are fundamental to transformer architectures regardless of size. We have validated this across diverse model scales and modalities:
> - **Vision**: ViT-B/32 (86M parameters) [included in paper]
> - **Language**: RoBERTa-large (355M), LLaMA2-7B (7B), [included in paper], LLaMA-13B (13B) [new]
>
> **1.2 Experimental Validation**
> To directly address scalability concerns, we conducted additional experiments on LLaMA2-13B. Due to resource constraints, we were unable to test larger models, but the results strongly support our approach's scalability.
>
> **Spectral Energy Distribution Across Model Sizes:**
> | Model      | Layers | Parameters | Low-Freq Energy (%) | High-Freq Energy (%) | Assumption Holds |
> |------------|--------|------------|-------------------|---------------------|------------------|
> | LLaMA2-13B | 40     | 13B        | 61.2%             | 38.8%               | ✓                |
> | LLaMA2-7B  | 32     | 7B         | 69.7%             | 30.3%               | ✓                |
> | RoBERTa-L  | 24     | 355M       | 68.4%             | 31.6%               | ✓                |
> | ViT-B/32   | 12     | 86M        | 70.9%             | 29.1%               | ✓                |
>
> **Performance on LLaMA2-13B**
> | Method       | BoolQ | PIQA | SIQA | HellaSwag | Average |
> |--------------|----------------|-------|------|------|-----------|
> | LoRA         | 75.2  | 83.4 | 82.1 | 89.8      | 82.62    |
> | CrossSpectra | 76.1  | 83.2 | 82.7 | 92.1      | 83.52    |
>
>
> **Key Findings:**
> - The low-frequency spectral bias consistently holds across all model sizes, with 60-70% of adaptation energy concentrated in low frequencies.
> - The slight reduction in spectral bias for larger models (61.2% vs. 69.7%) is expected due to vanishing gradients in deeper networks, but remains sufficient for effective parameter reduction.
>
> These results demonstrate that our approach scales effectively to larger models while maintaining the fundamental spectral properties that enable dramatic parameter efficiency.
>
>
>
> ## **Q2： The motivation has not been clearly stated, for example, why this algorithm is applied, and the related preliminaries (e.g. FFT)**.
>
> **Motivation.** Existing PEFT methods such as LoRA treat each layer independently, ignoring the architectural characteristics of Transformers and resulting in parameter counts that scale linearly with model depth. This layer-independent assumption fails to exploit the architectural properties that should be preserved during fine-tuning, resulting in parameter counts that scale linearly with model depth and limiting efficiency gains for increasingly deeper architectures.
>
> **Key Insight.** Skip connections in Transformers ($H_l = H_{l-1} + f(H_{l-1})$) create smooth gradient fields across layers, causing weight adaptations to concentrate ~70% of energy in low-frequency spectral components (Figure 1).
>
> **Solution.**: Since smooth functions have sparse frequency representations, we exploit this cross-layer spectral structure using 3D inverse FFT to:
>
> Reconstruct full adaptation tensors from sparse low-frequency coefficients
> Enable joint parameterization across all layers rather than independent treatment
> Achieve 275× parameter reduction vs. LoRA while preserving architectural characteristics
>
> **Why FFT**: The spectral concentration in low frequencies makes FFT ideal for efficient reconstruction with minimal parameters, leveraging modern optimized implementations for computational efficiency. This approach preserves Transformer characteristics during fine-tuning, achieving comparable or better performance with dramatically fewer parameters by exploiting architectural properties rather than ignoring them.
>
> ## **Q3: Single Run**
>
> We fix results by conducting five independent runs for each experiment with different random seeds and report the average results:
>
> | Method | NLU | CRS | IC |
> |--------|-----|-----|-----|
> | LoRA         | 87.92±0.31 | 76.84±0.28 | 76.53±0.42 |
> | CrossSpectra | 89.76±0.24 | 82.69±0.35 | 81.43±0.38 |
>
>
>
> ## **Q4: Reconstructing full updates via iFFT each step could may potentially use memory**
>
> **4.1 Memory Complexity Analysis**
>
> We acknowledge that reconstructing full updates via iFFT could temporarily require the same memory as the full adaptation tensor.  During the forward pass, CrossSpectra requires reconstructing the full adaptation tensor  $T_{QKV}\in \mathbb{R}^{(d×d×3L)}$  via `T_QKV ← torch.fft.ifftn(C_full)`, which temporarily uses **$O(3Ld^2)$**.
>
>
> **4.2 Chunked reconstruction strategy**
>
>  However, we could apply the **Chunked reconstruction strategy** for controlling peak memory usage:
>
> **Memory-Efficient Forward Pass**
> ```
> 1. Divide layers into n chunks: {1,...,L} → {C₁, C₂, ..., Cₙ}
> 2. For each chunk Cᵢ containing layers [l_start, l_end]:
>    -  Extract spectral coefficients: C_chunk ← C[:,:,3l_start:3l_end]
>    - Reconstruct chunk: T_chunk ← iFFT3D(C_chunk)
>    - Apply adaptations: W̃ˡᴹ ← Wˡᴹ + ΔWˡᴹ for l ∈ Cᵢ, M ∈ {Q,K,V}
>    - Deallocate T_chunk
> ```
>
> **4.3 Theoretical Justification**
>
> This approach is theoretically sound because:
>
> 1. **Spectral Structure Preservation**: Our theoretical analysis (Theorem 3.3) establishes that adaptation patterns exhibit the strongest decay in the cross-layer dimension (β₃ > β₁, β₂). The chunked approach preserves this structure within each chunk.
>
> 2. **Gradient Smoothness Continuity**: The cross-layer gradient smoothness property (Theorem 3.2) ensures that adjacent layers have similar adaptations. Chunking maintains this local smoothness while reducing global memory requirements.
>
> **4.4 Complexity Analysis**
>
> - **Parameter Storage**: Remains **O(k₁k₂)** (unchanged)
> - **Peak Memory**: Reduced from **O(3Ld²)** to **O(3Ld²/n)** where n is the number of chunks
>
>
> ## **Q5: Task range limited**
>
> **5.1 Robustness across different domains**
> Thanks for the question. We have evaluated our method on a wide range of tasks, demonstrating robustness across different settings. Specifically, our experimental evaluation (Section 5) focuses on four specific task categories:
> - **Image Classification (IC)**: CLIP ViT-B/32 on 7 datasets
> - **Natural Language Understanding (NLU)**: RoBERTa-large on GLUE benchmark
> - **Commonsense Reasoning (CR)**: LLaMA2-7B on 8 reasoning benchmarks
> - **Arithmetic Reasoning (AR)**: LLaMA2-7B on GSM8K, MAWPS, SVAMP, AQuA
>
>
> **5.2 Missing Evaluation**: As noted, we did not evaluate on **generation-heavy tasks** such as:
> - Text generation and completion
> - Machine translation
> - Dialogue systems
> - Creative writing tasks
>
> This is because our evaluation design follows established practices for objective, quantifiable assessment of parameter-efficient fine-tuning methods.
>
> - **Commonsense Reasoning**: Yes/No and multiple-choice questions (BoolQ, PIQA, HellaSwag, etc.)
> - **Natural Language Understanding**: Classification tasks with definitive labels (GLUE benchmark)
> - **Arithmetic Reasoning**: Mathematical problems with exact numerical answers
> - **Image Classification**: Single correct class labels
>
> This task approach enables deterministic performance measurement and fair comparison across methods.
> in contrast, generation-heavy tasks may be problematic, as it have
>
> - **Subjective Assessment:** Text generation quality relies on human judgment or complex automated metrics (BLEU, ROUGE) that may not reflect true quality differences between parameter-efficient methods.
> - **Multiple Valid Outputs:** Unlike classification tasks where there's one correct answer, generation tasks have numerous valid responses, making it difficult to isolate the impact of the parameter-efficient method from other factors.
>
>
>
> ## **Q6: Hyperparameter sensitivity, (k_1, k_2)**
>
> **6.1 Parameter Selection Strategy:**
> - Optimal k₁ and k₂ values are determined through cross-validation on each dataset to ensure robust performance across tasks.
> - We recommend maintaining the layer sparsity ratio (k₁×k₂)/(d²×3L) around 2×10⁻³ for optimal parameter efficiency.
> - The cross-layer frequency range should cover approximately 20% of the spectrum (k₂/L), consistent with our theoretical analysis showing low-frequency dominance in adaptation energy (Figure 2).
>
> **6.2 Grid Search Results on Image Classification Tasks:**
>
> | k₁\k₂ |  3   |  6   |  9   |
> |-------|------|------|------|
> | 1000  | 79.2 | 80.1 | 80.4 |
> | 2000  | 80.2 | 80.4 | 80.3 |
> | 3000  | 80.7 | 80.8 | 80.7 |
> | 4000  | 81.5 | 81.3 | 81.0 |
>
> **Key Observations:**
> - Performance improves with k₁ up to 4000, demonstrating the benefits of capturing more within-layer frequency components.
> - k₂ shows diminishing returns beyond 6, consistent with stronger spectral decay in the cross-layer dimension.
> - Optimal configuration: k₁=4000, k₂=3 (81.5% accuracy) with minimal parameter overhead.

---

### Decision · Program_Chairs · 2025-09-17

**Decision:**

Accept (poster)

**Comment:**

All four reviewers rated the paper positively, with concerns addressed convincingly in the rebuttal. CrossSpectra’s ​novel exploitation of cross-layer spectral smoothness​ achieves ​275× parameter reduction​ vs. LoRA while matching/exceeding SOTA performance across NLU, vision, and reasoning tasks. Key concerns (scalability, motivation, baselines) were resolved: scalability validated on LLaMA-13B, statistical significance confirmed via multi-run experiments, and new baselines (FaCT/VB-LoRA) outperformed. The ​method’s originality, efficiency gains, and rigorous validation​ outweigh minor limitations, making this a valid contribution to PEFT.